# miR-20a is upregulated in serum from domestic feline with *PKD1* mutation

**Marcela Correa Scalon**[1☯], **Christine Souza Martins**[1☯], **Gabriel Ginani Ferreira**[2‡], **Franciele Schlemmer**[2‡], **Ricardo Titze de Almeida**[2‡], **Giane Regina Paludo**[1☯¤]*

1 Veterinary Clinical Pathology Laboratory, College of Agronomy and Veterinary Medicine, University of Brasília, Brasília, Brazil, 2 Technology for Gene Therapy Laboratory, College of Agronomy and Veterinary Medicine, University of Brasília, Brasília, Brazil

☯ These authors contributed equally to this work.
¤ Current address: College of Agronomy and Veterinary Medicine, University of Brasilia, ICC Centro, Campus Universitário Darcy Ribeiro, Brasilia, DF, Brazil
‡ GGF, FS and RTA also contributed equally to this work.
* giane@unb.br

**Data Availability Statement:** All relevant data are within the paper and its Supporting Information files.

## Abstract

Polycystic kidney disease (PKD), also known as autosomal dominant polycystic kidney disease (ADPKD) is a genetically heterogeneous condition characterized by cysts in renal parenchyma. It is the most prevalent inherited disease of domestic cats. MicroRNAs (miR-NAs or ncRNA) are short, noncoding, single-stranded RNAs that may induce PKD cytogenesis by affecting numerous targets genes as well as by directly regulating PKD gene expression. We compared the relative expression profile of miR-20a, -192, -365, -15b-5p, and -16-5p from plasma and serum samples of nine domestic cats with *PKD1* mutation, detected by polymerase chain reaction (PCR), and a control group (n = 10). Blood samples from cats with *PKD1* mutation provide similar concentrations of microRNAs either from plasma or serum. Serum miR-20a is upregulated in PKD group with p < 0.005; Roc curve analysis showed an AUC of 90,1% with a cut-off value sensitivity of 77.8% and specificity of 100%. This data provides important information regarding renal miRNA expression in peripheral blood sampling.

## Introduction

Polycystic kidney disease (PKD) or autosomal dominant polycystic kidney disease (ADPKD) is a genetic condition that causes the formation of cysts in renal parenchyma. In cats it is the most prevalent genetic disease affecting mainly Persian and Persian related breeds. It is also responsible for the develop of chronic kidney failure in elderly cats [1, 2]. Global prevalence of PKD in cats is about 30–38% [3], with 37–49% in Persian or Persian related cats. Furthermore, this condition can be detected in American Shorthair, British Shorthair, Sottish Folds and mixed-breed cats [3, 4]. There are four distinct types of renal cystic diseases. Potter syndrome type I is referred to as autosomal recessive polycystic disease (ARPKD), type II as renal cystic dysplasia, type III as autosomal dominant polycystic kidney disease (ADPKD), and type IV

**Funding:** The author(s) received no specific funding for this work.

**Competing interests:** The authors have declared that no competing interests exist.

occurs when a longstanding obstruction in either the kidney ureter leads to cystic kidneys or hydronephrosis [5]. The ADPKD is caused by mutation in either one of two principal genes, PKD1 and PKD2. In humans and domestic cats, PKD1 mutation is the most common manifestation, present in ~80% of the carriers' population. It is known that microRNAs can affect the development of PKD in direct or indirect ways [6].

Renal cysts are genetically heterogeneous conditions and both PKD1 and PKD2 genes from dominant form, can be inherited in a recessive way, known as autosomal recessive polycystic kidney disease (ARPKD). There are an increasing number of genes that have to be considered in cystic kidney diseases showing that genetic analysis of different target genes increases detection rate and allows better interpretation on genetic diagnoses [5]. Similar to humans, domestic cats have genetic diseases and some of them are well known as PKD and hypertrophic cardiomyopathy (HCM) [1, 7].

Illness comparative studies are very important in genetic inheritance, especially the ones that involve humans and animal models, where diseases profiles are similar as PKD [3]. In a study that made the first analysis for microRNAs expression from renal tissue in dogs and cats, a total of 277, and 276 microRNAs were detected in cat renal cortex and medulla, respectively [8]. Another study compared RT-rtPCR quantification of five microRNAs (miR-192, -20a, -365, -15b-5p and -16-5p) in plasma and serum from domestic cats [9]. These set of microRNAs shows differential patterns of expression during physiologic and pathologic conditions of kidney tissue in dogs and cats [8].

In mice, the progression of renal fibrosis has been correlated with an increased expression of miR-192, which is increased by TGF-β in several human and mouse renal cell types. miR-192 is strongly associate with renal diabetic conditions and it is not directly correlated to ADPKD. Also for miR-192, it observed a progressive decrease in its expression following renal ischemia reperfusion injury (IRI), most likely reflecting contraction of the kidney, which also mirrored changes observed in TGF-β expression [10, 11].

miR-20a was overexpressed after IRI in a mouse model of ischemic injury. The up-regulation of miR-20a suggests that expression of this miRNA may prevent apoptosis resulting from IRI and induce proliferation to promote repair [11]. The miR-20a belongs to the miR-17/92 cluster and is related to ADPKD and IRI. The over-expression of miR-20a reduces apoptosis, proliferation arrest and the release of inflammatory cytokines, thus, it may play a role as a protective molecule in the pathogenesis and progression of acute kidney injuries [12].

In humans, miR-365 was first demonstrated to target on the 3'UTR of PKHD1gene specially and repressed the expression of PKHD1 at post-transcriptional level. Further study revealed that miR-365 post-transcriptionally modulated-PKHD1 inhibited cell–cell adhesion in part through E-cadherin. The bioinformatics analysis of the entire 3'UTR of human PKHD1 gene demonstrated that this region was highly conserved across different species, and that miR15-a and miR-17 play roles in *PKD1* and *PKD2* genes expression [13]. miR-15b-5p and miR-16-5p are regulators of cell proliferation and apoptosis in rats, they are part of miR-15a/16-1 and miR-15b/16-2 clusters that regulates embryonic pattering in a variety of cell lines and are not directly correlated to ADPKD [14].

Due to the lack of information on the expression of miRNAs in domestic felines, especially around the PKD disease development, the aim of this study was to analyze the expression rate of five miRNAs (miR-20a, -192, -365, -15b-5p, and -16-5p) in plasma and serum samples, from healthy cats and PKD carriers. At first, the develop of a working protocol to detect the miRNAs was challenging. But after testing the samples, all five targets were available to be analyzed, with the possibility to represent new biomarkers candidates for renal diseases processes.

## Material and methods

In this study, we compared the relative expression of five microRNA targets (miR-192, -20a, -365, -15b-5p and -16-5p) in blood samples between a control group (10 healthy cats) and the PKD carrier group (9 clinically healthy cats). The selected animals were client-owned domestic cats. All owners were informed about the study and agreed to sign a consent form. This document allowed the veterinarian team to collect blood and urine samples from the cat, and also provide the necessary information for both the owner (about the study) and the researchers (about the cat) S1 File. Control group had six males and four females, one Persian and nine mixed breeds, with ages of seven months to seven years [9]. PKD carrier group had five males and four females, seven Persian and two mixed breeds, with ages of two to 11 years. Blood samples were collected by venipuncture of cephalic or jugular veins into K3EDTA-containing tubes for complete blood cell count (CBC), *PKD1* mutation gene PCR, and for plasma separation for miRNA extraction. The tubes with non-additives were used for serum with the purpose to biochemical analysis, and serum miRNA extraction. As complementary tests, urine samples collected by cystocentesis were used for urinalysis. CBC was performed in an automatic cell counter (ABX Micros ESV 60, Horiba). Biochemical analysis included blood urea nitrogen (BUN), creatinine, total serum protein, albumin, phosphorus (P) and potassium (K) was performed in a biochemical analyzer (Cobas C111, Roche), and SDMA by Idexx Laboratories (S1–S3 Tables).

All animals were tested for PKD PCR using Illustra Blood genomicPrep Mini Spin Kit (GE Healthcare) for DNA extraction. DNA samples were stored at -20˚C prior to use. All amplifications were performed in duplicate using C1000 thermal cycler (Bio-Rad). Purified samples were tested for glyceraldehyde-3-phosphate dehydrogenase (GAPDH) to verify DNA quality, integrity, and the presence of PCR inhibitors using GAPDH-F (5' - CCTTCATTGACCT-CAACTACAT -3') and GAPDH-R (5' - CCAAAGTTGTCATGGATGACC -3') primers [15]. After GAPDH confirmation, samples were tested for *PKD1 mutant* gene using PKD-F3 (5' - AGAGGCAGACGAGGAGCACT -3') and PKD-R2 (5' - GCCTCGTGGAGAAGGAGGT -3') primers as previously described [4]. All cats from PKD group tested positive for *PKD1* mutation PCR.

The procedures to obtain plasma for microRNA analysis used blood in K3EDTA-containing tubes. Plasma was separated from blood cells within 1 h of collection, at $1,300 \times g$ for 10 min in a 4˚C refrigerated centrifuge for removal of formed elements. A second centrifugation was performed at $3,000 \times g$ for 10 min at 4˚C to remove any residual platelets [16]. Non-additives tubes were centrifuged at $1,814 \times g$ for 5 min to obtain serum, that was carefully aspirated to avoid contamination with cellular debris. Samples were processed at the Veterinary Clinical Pathology and Technology for Gene Therapy Laboratories, at the College of Agronomy and Veterinary Medicine, University of Brasilia, Brazil. The Animal Use Ethics Committee, CEUA-UnB—protocol 82/2018 approved our experimental methods, including the Consent Form (S1 File) signed by the cats owners.

The samples were separated into 100-μL aliquots to minimize the effects of repeated cycles of thawing and freezing, given that these temperature changes reduce the number of miRNA molecules available [17], and frozen at −80˚C until the microRNA extraction procedure. Extraction was performed using miRNeasy mini kit (Qiagen) following the manufacturer's recommendations, with addition of a supplementary protocol [18]. As an extraction and normalizing control, 3.5 μL of the synthetic cel-miR-39-3p (miRNeasy serum/plasma spiked-in control; Qiagen) was spiked into each sample at a concentration of $1.6 \times 10^8$ copies/μL [19]. Final elution volumes in RNase-free water were 50 μL as recommended by manufacturer.

Fluorometric quantification of miRNA was done using 2 μL of extracted sample (Qubit 2.0, microRNA assay kits; Life Technologies) following the manufacturer's recommendations. The

miRNA concentrations were 0.25–1.0 ng/μL. All samples were diluted to 0.25 ng/μL to standardize the miRNA concentration for the following RT-rtPCR protocol.

The TaqMan RT kit (Thermo Fisher) was used for complementary DNA synthesis, followed by real-time PCR (rtPCR). Stem-loop RT primers were used for the quantification of the miRNAs (Table 1) [9]. Preparation of the reagent mix followed the manufacturer's instructions (TaqMan; Life Technologies). The rtPCR assays were performed using a QuantStudio 12K (Life Technologies) thermocycler. All samples were tested in triplicate and all assays included RNasefree water as a negative control [20].

For the first RT-PCR reaction, the miRNA template was converted into a complementary DNA (cDNA) using a reverse transcriptase (RT) enzyme. The cDNA was then used as a template for exponential amplification using rt-PCR [21–23]. The relative quantification involved the co-amplification of a synthetic spike-in miRNA control (cel-miR-39-3p) simultaneously with the target of interest. All samples, including the synthetic control, were amplified in triplicates. Then, the Ct values were average to calculate normalized delta Ct ($\Delta Ct$ = Ct (miR of interest)–Ct (cel-miR-39-3p). Once normalized, a $\Delta Ct$ control average was made using the $\Delta Ct$ values from the control group. The next step, the $\Delta\Delta Ct$ for each miRNA calculation was made with the following formula: $\Delta\Delta Ct$ = $\Delta Ct$ (miR of interest)–$\Delta Ct$ (Control average). The results of the analysis were expressed as the ratios of the target signal to the spike-in synthetic control signal. To get the relative expression of the target, the formula needs to do 2 to the power of negative $\Delta\Delta Ct$ [Fold change = $2^{\wedge}-(\Delta\Delta Ct)$ or $2^{-\Delta\Delta Ct}$], given the final fold-change value (Figs 1 and 2), which can then be used for the comparison between the samples in the estimation of relative miRNA expressions. All fold change values were log transformed for consistent statistical analysis [20, 24, 25].

*Shapiro-Wilk* test was used to analyze normal distribution. An unpaired student t test was used for comparisons between data with normal distribution. *Mann-Whitney* test was used as a nonparametric test. The level of significance was $p \leq 0.05$. ROC curve graphics were created by plotting the true positive rate (sensitivity %) against the false positive rate (specificity %) to generate the cumulative distribution function (area under the curve–AUC). Confidence interval (CI) was 95% and the level of significance was $p \leq 0.005$. To estimate the probability of the cut-off value, Youden's index was used. All analyses were performed using Prism 6 (GraphPad).

## Results and discussion

Our method successfully amplified the targets (miR-192, miR-365, miR-20a, miR-16-5p, and miR-15b-5p) within different patterns of expression. All of them can be detected both in plasma or serum from domestic cat blood samples (Figs 1 and 2). As a descriptive data, Table 2 shows that serum have more miR-20a (p = 0.0011) and miR-16-5p (p = 0.0003) than plasma. These data are consistent with Scalon et al, 2021 results, where they conclude that serum can provide more miRNA than plasma, but both samples are viable sources for miRNA measurement in domestic cat. Tables 3 and 4 shows Cts descriptive data from PKD and Control group respectively, with $\Delta Ct$ *p* value calculated separately for each of them comparing plasma versus serum relative expression.

For PKD group there was no significant difference between plasma and serum miRNAs, therefore, blood samples from cats with *PKD1* mutation provides the same relative expression of miRNAs both in plasma and in serum. For Control group, serum samples have significantly more miRNA for miR-20a (p < 0.0001) and miR-16-5p (p = 0.0002). As previously described, samples may show more microRNAs from blood cell rupture during coagulation, which could be a disadvantage, but this type of sample can be very stable even under denaturation and

**Table 1. microRNAs, stem-loop RT primers, mature microRNA sequences, and physiologic or pathologic correlation [8].**

| microRNA | Stem-loop RT primers | Mature microRNA sequence | Physiologic or pathologic correlation |
|---|---|---|---|
| miR-192-5p | hsa-mir-192: GCCGAGACCGAGUGCACAGGGCUCUGACCUUAUGAAUUGACAGCCAGUGCUCUCGCUCUCCCUCUGGCUGCCAAUUCCAUAGGUCACAGUAUGUUCGCCUCAAUGCCAGC | CUGACCUAUGAAUUGACAGCC | Renal fibrosis |
| miR-20a-5p | hsa-mir-20a: GUAGCACUAAAGUGCUUAUAGUGCAGGUAGUGUUUAGUUAUCUACUGCAUUAUGAGCACUUAAAGUACUGC | UAAAGUGCUUAUAGUGCAGGUAG | Ischemic reperfusion injury |
| miR-365-3p | hsa-mir-365a: ACCGCAGGGAAAAAUGAGGGACUUUUGGGGGCAGAUGUGUUUCCAUUCCACUAUAAUGCCCUAAAAAUCCUUAUUUGCUCUUGCA | UAAUGCCCUAAAAAUCCUUAU | Polycystic kidney disease and hepatic disease 1 |
| | hsa-mir-365b: AGAGUGUUCAAGGACAGCAAGAAAAAUGAGGGACUUUCAGGGGCAGCUGUGUUUCUGACUCAGUCAUAAAUCCUUAUAUUGUUCUUGCAGUGUGCAUCGGG | | |
| miR-15b-5p | hsa-mir-15b: UUGAGGCCUUAAAGUACUGUAGCAGCACAUCAUGGUUUACAUGCUACAGUCAAGAUGCGAAUCACUAUUUGCUGUCUAGAAAUUAAGGAAAUUCAU | UAGCAGCACAUCAUGGUUUACA | Proliferation and apoptosis regulation |
| miR-16-5p | hsa-mir-16-1: GUCAGCAGUGCCUUAGCAGCACGUAAAUAUUGGCGUAAAUAUUGGCGUAAGAUUCUAAAAUUAUCUCCAGUAUUAACUGUGCUGCUGAAGUAAGGUUGAC | UAGCAGCACGUAAAUAUUGGCG | Proliferation and apoptosis regulation |
| | hsa-mir-16-2: GUUCCACUCUAGCAGCACGUAAAUAUUGGCGUAGUGAAAUAUAUUAAACACCAAUAUUACUGUGCUGCUUUAGUGUGAC | | |

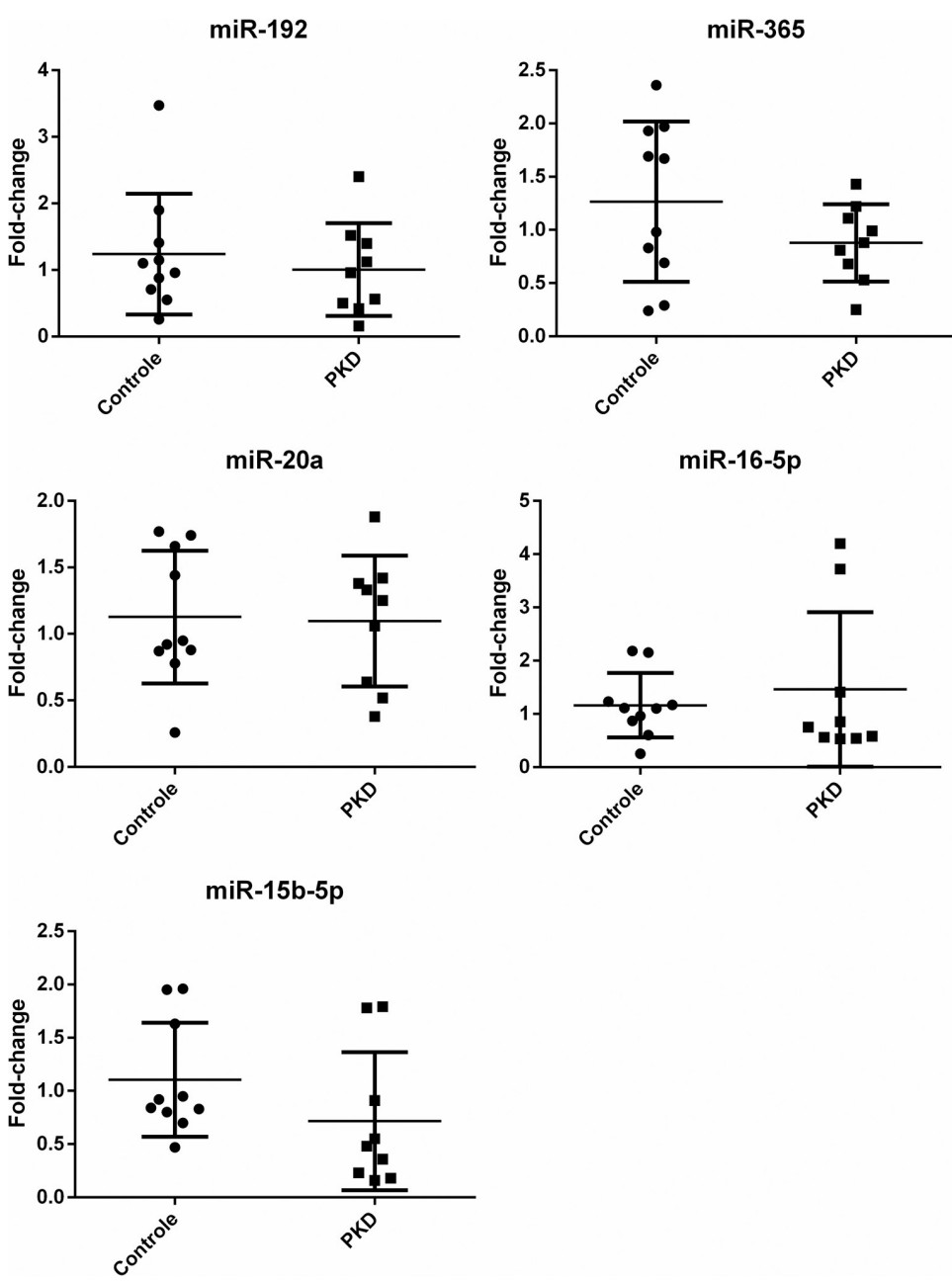

**Fig 1. Fold change of plasma microRNAs (miR-20a, miR-192, miR-365, miR-16-5p, miR-15b5p) from domestic cat.** RT-rtPCR assays. Normalized with cel-miR-39-3p (p > 0.05).

degradation conditions. Even with plasma viability, the use of serum samples is easier to handle from collection to processing in the laboratory [9, 26, 27].

Despite plasma miR-192, -365, -20a, and -15b-5p were downregulated, and miR-16-5p was upregulated in PKD group, relative expression did not differ (p > 0.05) between groups (Table 5). These results were different in our samples when comparing the relative expression for these targets in other studies using mice, rat and/or humans renal tissue [10–14]. Future studies should use more samples and analyze the expression pattern in different tissues and blood samples.

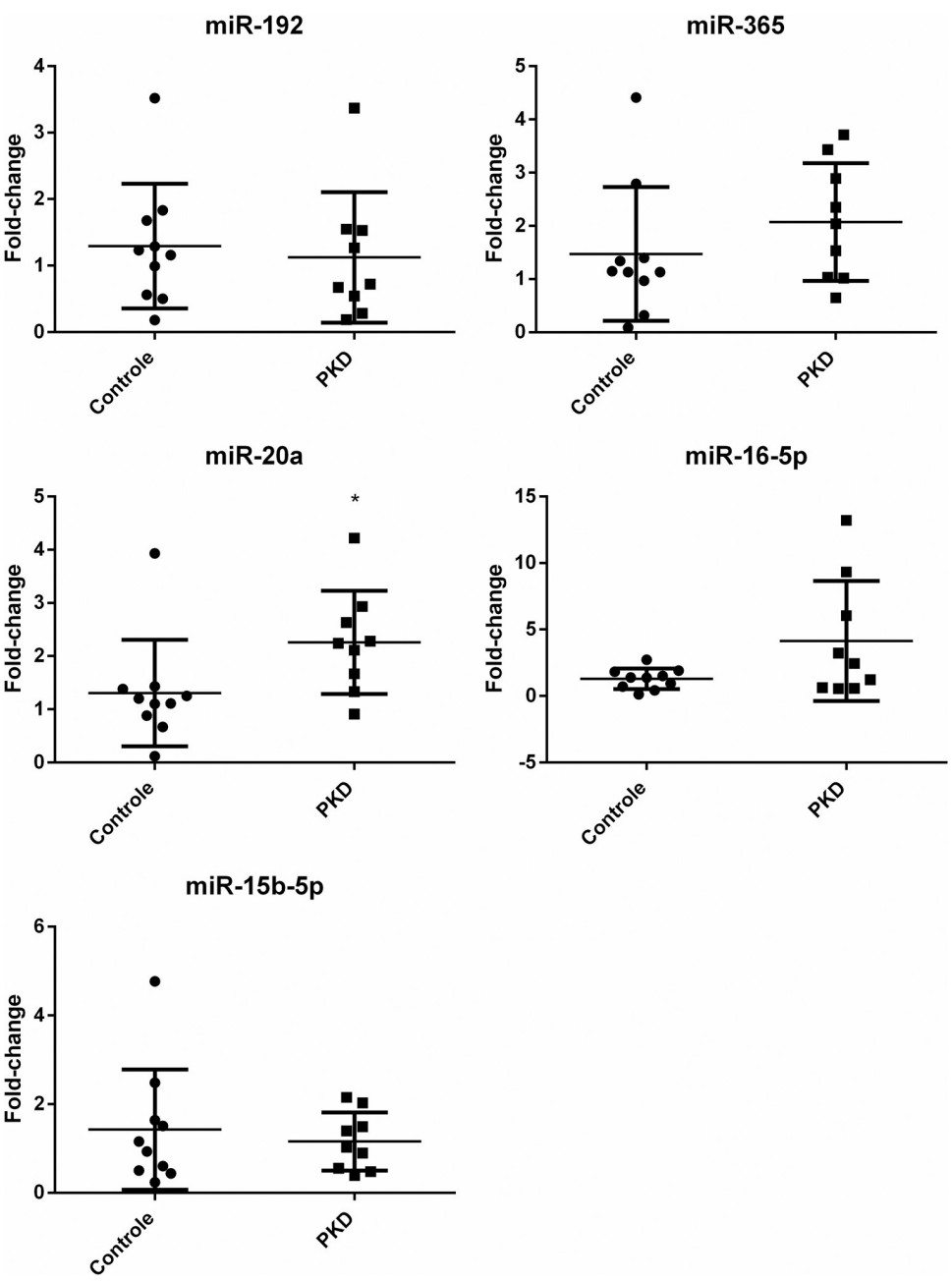

**Fig 2. Fold change of serum microRNAs (miR-20a, miR-192, miR-365, miR-16-5p, miR-15b-5p) from domestic cat.** RT-rtPCR assays. Normalized with cel-miR-39-3p. miR-20a is upregulated in PKD group *(p < 0.005), all the others *p* values were p > 0.05.

Compared to the Control group, serum miR-20a was upregulated in PKD group (Table 6, p<0.005), and for plasma it was downregulated but no statistical difference. The use of serum, in this case is preferable but it is very important to consider the difference between the type of samples. Plasma is more complex to handle and will mostly provide less miRNA [9, 16], this can explain why our serum samples showed a different pattern for miR-20a. This target is related to ischemic reperfusion injury (IRI) and autosomal dominant polycystic kidney disease [11, 12]. IRI is often associated with microvascular injury that lead to an increase of diffusion

**Table 2. Cycle threshold values (mean ± SEM) for microRNA derived from feline plasma and serum samples of all groups (control + PKD).**

| miRNA | Plasma | Serum | ΔCt *p* value |
|---|---|---|---|
| miR-192 | 29.9 ± 0.3 | 30.4 ± 0.3 | 0.6172 [§] |
| miR-20a | 27.0 ± 0.3 | 28.4 ± 0.2 | 0.0011 [§] |
| miR-365 | 29.5 ± 0.4 | 30.2 ± 0.2 | 0.0608 [§§] |
| miR-15b-5p | 31.1 ± 0.4 | 31.2 ± 0.3 | 0.6892 [§] |
| miR-16-5p | 22.1 ± 0.3 | 24.1 ± 0.4 | 0.0003 [§] |

[§] unpaired Student's t-test

[§§] Mann Whitney test

and fluid filtration across the tissues [28]. This information in cats with *PKD1* mutation may be associated with the mechanism of cyst formation as that *PKD1* deficient cells fail to switch off the normal renal injury-induced repair program. Instead, they continue to proliferate, resulting in cyst formation. It is also important to consider that, in different stages of PKD, the levels of miRNAs associated may vary both in renal tissue and serum. Besides that, there is also the genetic hypothesis for a 'two-hit' or 'three-hit' PKD model, that consider different progressions for cystogenesis in early and adult life in humans [29].

Although they are not clearly understood, cysts can be involved with improper functioning of several pathways including cell proliferation, apoptosis, cell polarity and fluid secretion [30]. Maybe, due to the upregulation of miR-20a observed on this study, the cysts formation could be related to an ischemic injury that can possibly be detected in serum. It is known that reperfusion injury is correlated with overexpression of miR-20a that modulates translation of E2F transcription factors, which regulate proliferation and apoptosis in renal tissue in mice [11, 31]. A recent *in vitro* study investigated function and mechanisms of miR-20a in acute kidney injury. They describe that overexpression of miR-20a promoted cell viability, inhibited apoptosis rate and inhibited the expression of apoptotic factors and inflammatory cytokines in HK-2 cells after LPS stimulation. In addition, CXCL12 was identified as a direct target of miR-20a by luciferase reporter gene assay, and CXCL12 expression was negatively regulated by miR-20a [12]. Something similar may occur in cats, as this miRNA was predominantly expressed in renal medulla [8].

Our cat samples showed expression fold-change patterns but no statistical difference in miR-192, -365, -16-5p, and -15-5p (Tables 5 and 6). Also related to IRI, miR-192 is down-regulated in mice tissue [11] and is overexpressed in some diabetic kidney disease in mouse and humans [10]. These results possibly happen because of IRI, as we have miR-20a expression pattern related with the same pathophysiologic condition, besides others physiologic cell

**Table 3. Cycle threshold values (mean ± SEM) for microRNA derived from feline plasma and serum samples of the PKD group.**

| miRNA | Plasma | Serum | ΔCt *p* value [§] |
|---|---|---|---|
| miR-192 | 30.7 ± 0.4 | 30.9 ± 0.4 | 0.7855 |
| miR-20a | 27.6 ± 0.5 | 28.1 ± 0.4 | 0.1407 |
| miR-365 | 30.2 ± 0.4 | 30.3 ± 0.3 | 0.7694 |
| miR-15b-5p | 32.1 ± 0.6 | 31.5 ± 0.5 | 0.2276 |
| miR-16-5p | 22.5 ± 0.5 | 23.7 ± 0.6 | 0.1174 |

[§] ΔCt *p* value calculated by unpaired Student's t-test

**Table 4. Cycle threshold values (mean ± SEM) for microRNA derived from feline plasma and serum samples of the control group.**

| miRNA | Plasma | Serum | ΔCt *p* value [§] |
|---|---|---|---|
| miR-192 | 29.2 ± 0.4 | 29.9 ± 0.4 | 0.6727 |
| miR-20a | 26.4 ± 0.4 | 28.5 ± 0.5 | < 0.0001 |
| miR-365 | 28.8 ± 0.5 | 30.5 ± 0.6 | 0.0506 |
| miR-15b-5p | 30.2 ± 0.3 | 30.9 ± 0.5 | 0.8108 |
| miR-16-5p | 21.7 ± 0.4 | 24.3 ± 0.6 | 0.0002 |

[§] ΔCt *p* value calculated by unpaired Student's t-test

functions [10, 11]. Serum miR-365 is upregulated in PKD group. In humans, this miRNA is related to autosomal recessive polycystic kidney disease (ARPKD) and *PKHD1* gene, that was correlated with cell-cell adhesion in part through E-cadherin. The 3'UTR of this gene is highly conserved in evolution [13], but it not sequenced in domestic cats. Nonetheless, the miR-365 is present in tissue and blood samples in this specie [8, 9] and is possibly an interesting issue to be discussed in future researches. miR-15 and -16 clusters are highly conserved among mammalian species. The expression pattern of individual miRNAs spliced from miR-15a/16-1 and miR15b/16-2 clusters can be found in various tissues as heart, brain, lung, kidney, and small intestine [14]. In addition to that, our results showed that miR-15b-5p and miR-16-5p can be measured as circulating targets in cat plasma and serum. miR-15b-5p is downregulated, and miR-16-5p is upregulated and it is known that they play very important roles in regulating cell proliferation and apoptosis by targeting cell cycle proteins [32, 33]. The investigation of the biological functions of those targets, using circulating and tissue measurements, can provide new clues to understanding their implication in renal disease future studies. But it is important to highlight that they are not good targets to compare changes in plasma and serum following PKD in cats.

Kidney structures are complexly organized with essential contributions from various cell types. Therefore, comprehensive microRNA expression data are quite important to help clarify the function of microRNAs and to identify the ones that could be successful used as biomarkers [34], such as the miR-20a that is upregulated in serum samples from cats with *PKD1* mutation gene. It is also important to consider that, as for PKD in humans, the cat study model indicates that modifying genes, epigenetic mechanisms, and stochastic and/or environmental factors considerably influence the clinical course in the development of kidney diseases [1, 4, 5]. In this study it was possible to detect the presence and relative expression of miR-192, -365, -20a, -16-5p, and -15b-5p as circulating particles. However, the concentration of miRNAs would probably differ when comparing circulating and tissue relative expression.

Considering that the relative expression of miRNAs suggests a specific pattern for plasma and serum samples, a study with more individuals can benefit future understanding on

**Table 5. Comparison between control and PKD group on miRNA expression ratios from feline plasma.**

| Plasma microRNA | Fold change | Expression | *p* value |
|---|---|---|---|
| miR-192 | 0.81 | Down | 0.618 |
| miR-365 | 0.70 | Down | 0.179 |
| miR-20a | 0.97 | Down | 0.892 |
| miR-16-5p | 1.26 | Up | 0.486 |
| miR-15b-5p | 0.64 | Down | 0.078 |

**Table 6. Comparison between control and PKD group on miRNA expression ratios from feline serum.**

| Serum microRNA | Fold change | Expression | *p* value |
|---|---|---|---|
| miR-192 | 0.87 | Down | 0.705 |
| miR-365 | 1.40 | Up | 0.268 |
| miR-20a | 1.68 | Up | <0.005* |
| miR-16-5p | 3.18 | Up | 0.275 |
| miR-15b-5p | 0.81 | Down | 0.757 |

miRNA targets. In this study, the microRNAs were chosen as potential biomarkers for renal diseases and miR-20a is probably a useful target in serum for biomarker studies considering PKD in cats. It has been proposed that a single miRNA can target more than hundred genes and one gene can be target of several miRNAs, and that each one are correlated also with various physiologic cell processes [35, 36]. These non-coding RNA particles have important regulatory roles in a board range of biological processes, including the pathogenesis of malignancy, cellular differentiation, proliferation apoptosis, and gene regulation [6].

For individuals with *PKD1* mutation gene, it is already known that the cysts formation is expected, but this information does not show the evolution of renal tissue injure. Also, the relative expression pattern of miRNAs, combined with others clinical findings, can be an important information on prognostic and monitoring the patients [5]. Our results shows that miR-20a, miR-192, miR-365, miR-16-5p, miR-15b-5p are present as circulating particles, and, other studies shows that they are also expressed in renal tissue [8, 10–14]. The presence of cysts is a condition that is correlated to a genetic mutation, and, for that reason, can provide insight into the regulation of miRNAs. The use of a transgenic or knockout/knockdown model for PKD disease studies can open a new pathway for monitoring and controlling the mechanisms of cyst formation as well as the use of circulating miRNAs measurements that can be of great help.

Our analysis showed that there was no difference for serum miR-192, -365, -16-5p, -15b-5p, and all plasma microRNAs Roc curve analysis. The ROC curve is a simple and intuitive statistic test that can discriminate healthy and not healthy individuals and we decided to compare both groups as a way of predict a possibly test result considering circulating miRNA measurements. The AUC discriminatory power of 0.5 suggests no discrimination, 0.7 to 0.8 is considered acceptable, 0.8 to 0.9 is considered excellent, and more than 0.9 is considered outstanding [37]. Figs 3 and 4 shows all the ROC curves analysis from targets microRNAs between the healthy control group and the PKD group.

For serum miR-20a, the cut-off value is 1.55 with sensitivity of 77.8% and specificity of 100%. Considering this result, it is an excellent parameter with significant statistical difference for relative expression of serum miR-20a between healthy control group and PKD group. An interpretation of this difference is probably explained by the presence of cysts in renal parenchyma and consequently upregulated expression of the target in serum.

For diagnostic purposes, PKD can be identified by PCR of the *PKD1* mutant gene. However, there are other mutations in different exons of the *PKD1* gene. In humans it is known that various germline mutations for ADPKD and ARPKD currently listed in the ADPKD mutation database are predicted to be of a protein truncating character [5]. As human and domestic cat models for genetically renal disease are similar, it can be expected that such circumstance may be seen in cats [1, 4]. Therefore, the pattern of miR-20a relative expression can help on monitoring cyst formation. It might be helpful to also distinguish between acquired and inherited forms of renal cysts, so that patients with no genetic mutation could present a microRNA relative expression pattern if cysts exist.

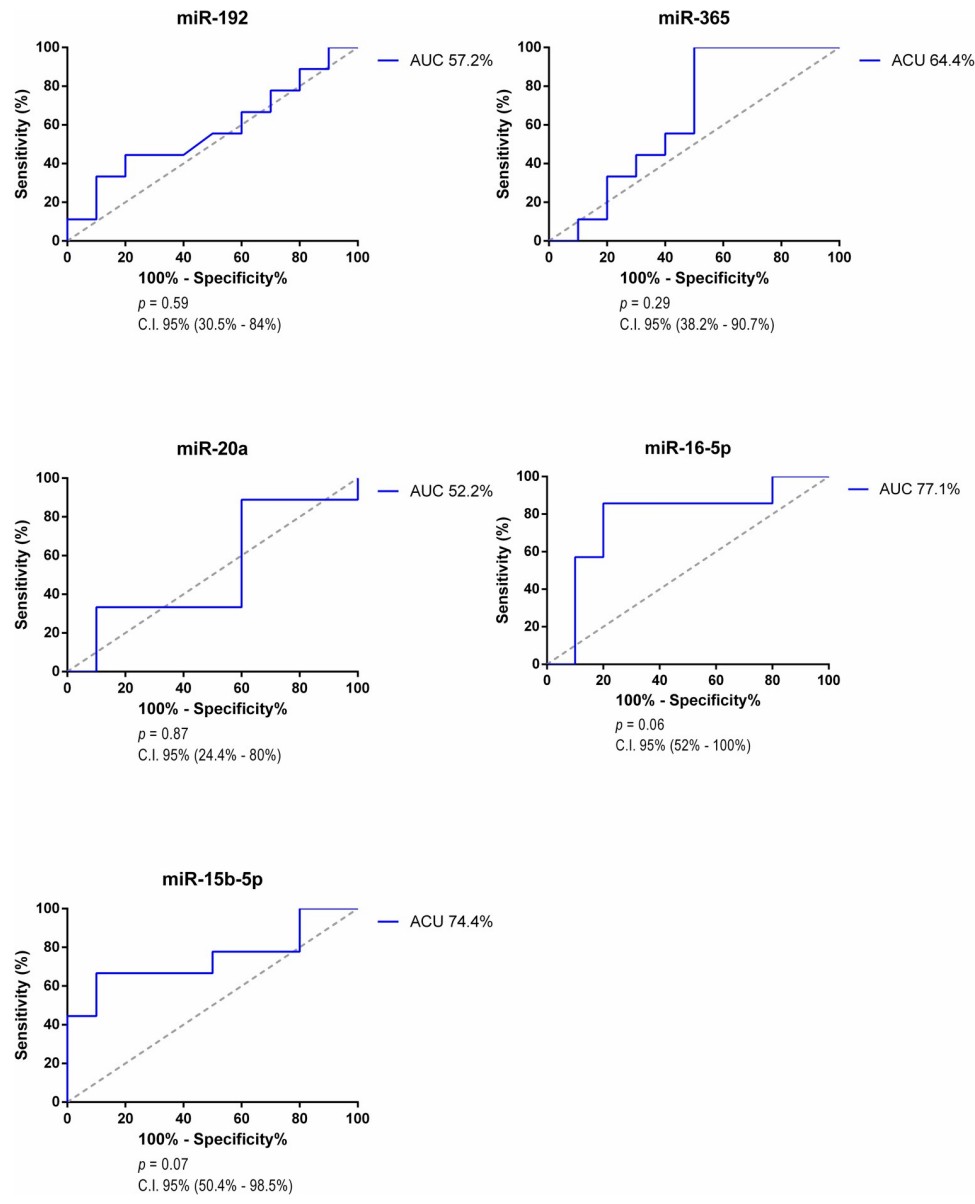

**Fig 3. ROC curve of plasma microRNAs (Control x PKD).**

Since renal injury features differ among kidney diseases, selected microRNA targets needs to be carefully considered and compared. Each kidney disease depends on the type of cells expressing the microRNA, in particularly for application of these molecules as disease markers that can be monitored by blood sample tests. For the analysis of miRNAs associated with renal pathogenesis, identification of disease type-specific miRNAs and their expressing patterns are important because the kidneys are structurally and functionally remarkably complex organs. Even not directly related to ADPKD, some miRNAs detected in renal tissue appear to be expressed in serum and plasma, probably by events of regulating apoptosis, proliferation, and inflammation processes. Measuring them may point us toward new pathways that can be targeted to prevent or modulate renal injury.

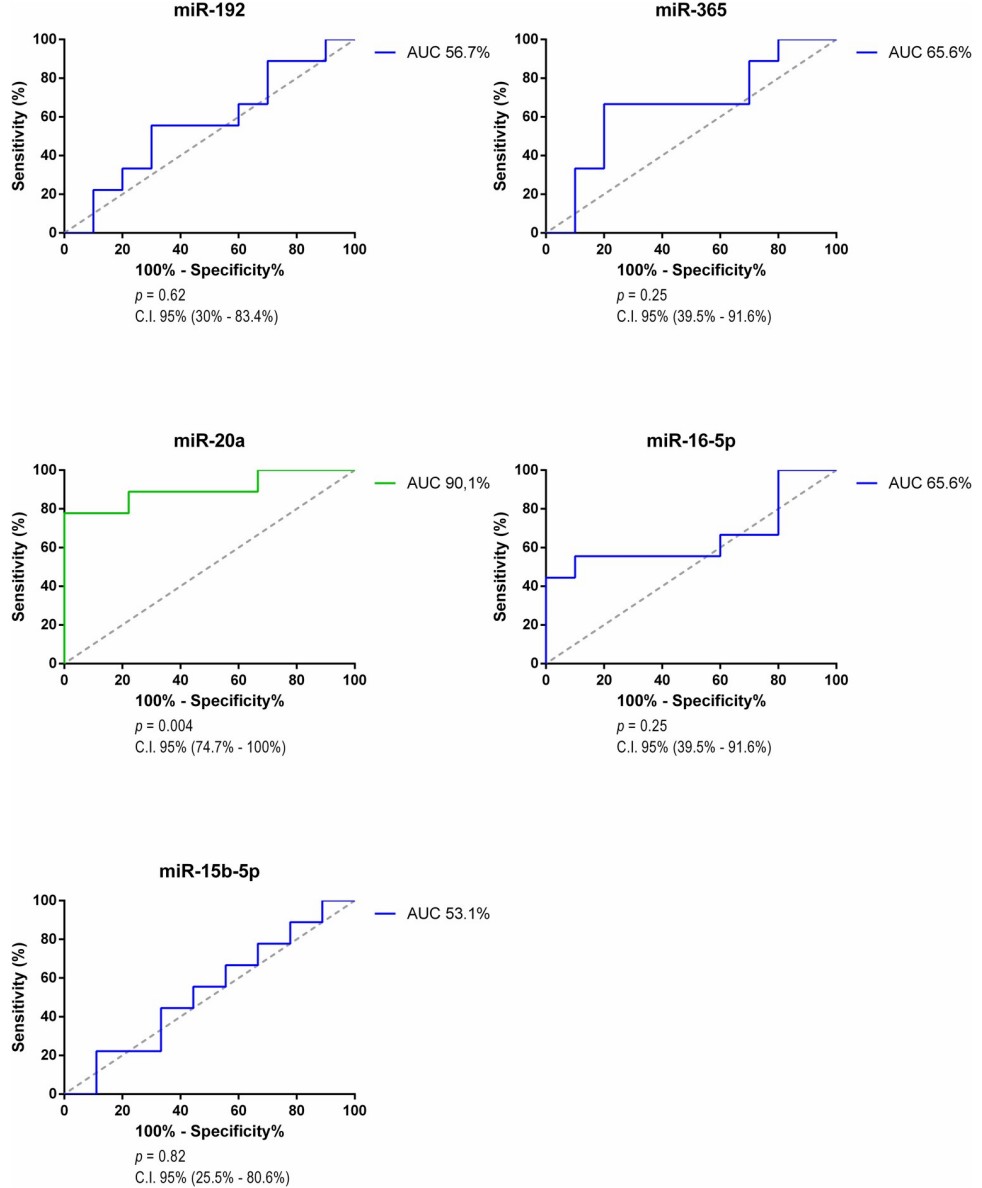

**Fig 4. ROC curve of serum microRNAs (Control x PKD).**

## Conclusions

For PKD cat samples, serum miR-20a is upregulated and is a potential biomarker target for detecting the presence of renal cysts, with a sensitivity of 77.8% and specificity of 100%. There was no difference between plasma and serum measurements of miR-192, -365, -15b-5p, and -16-5p. For that reason, we consider working with serum a better choice, mostly for laboratory handling and sample preservation. Further data using bigger cohorts would be helpful to evaluate experimental results in kidney disease pathways for cats and humans.

The set of microRNAs chosen was from a previously study that used kidney tissues and the HiSeq 2000 platform, comparing dog and cat samples with data on miRBase (http://www.mirbase.org/) [8]. Our analysis detected the same microRNA in plasma and serum samples from domestic cats. This information may help future studies on molecular pathogenesis of

renal cystic diseases in cats, using blood samples instead of organ tissues. Recently, miRNAs have been adopted not only as disease markers but also as therapeutic targets [34].

## Supporting information

**S1 File. Research informed consent form (English and Portuguese).**
(PDF)

**S1 Protocol. Extraction protocol for miRNA.**
(PDF)

**S2 Protocol. Quantification protocol for miRNA.**
(PDF)

**S3 Protocol. Reverse transcription protocol for miRNA.**
(PDF)

**S4 Protocol. Real time PCR protocol for miRNA.**
(PDF)

**S1 Table. Mean, standard deviation (SD) and standard error of the mean (SE) from clinical parameters (blood count, biochemical and urinalysis) of control group.** Where Hematocrit (HCT), Red Blood Cell (RBC), Hemoglobin (HGB), Mean corpuscular volume (MCV), Mean corpuscular hemoglobin concentration (MCHC), White blood cell (WBC), Blood Urea Nitrogen (BUN), Phosphorus (P), Potassium (K), Symmetric dimethilarginine (SDMA), Urine Protein/Creatinine Ratio (UPCR), Gamma-Glutamyl Transpeptidase (GGT).
(PDF)

**S2 Table. Mean, standard deviation (SD) and standard error of the mean (SE) from clinical parameters (blood count, biochemical and urinalysis) of PKD group.** Where Hematocrit (HCT), Red Blood Cell (RBC), Hemoglobin (HGB), Mean corpuscular volume (MCV), Mean corpuscular hemoglobin concentration (MCHC), White blood cell (WBC), Blood Urea Nitrogen (BUN), Phosphorus (P), Potassium (K), Symmetric dimethilarginine (SDMA), Urine Protein/Creatinine Ratio (UPCR), Gamma-Glutamyl Transpeptidase (GGT).
(PDF)

**S3 Table. Clinical parameters (blood count, biochemical and urinalysis) of individual samples.** Hematocrit (HCT), Red Blood Cell (RBC), Hemoglobin (HGB), Mean corpuscular volume (MCV), Mean corpuscular hemoglobin concentration (MCHC), White blood cell (WBC), Blood Urea Nitrogen (BUN), Phosphorus (P), Potassium (K), Symmetric dimethilarginine (SDMA), Urine Protein/Creatinine Ratio (UPCR), Gamma-Glutamyl Transpeptidase (GGT).
(PDF)

## Author Contributions

**Conceptualization:** Gabriel Ginani Ferreira, Franciele Schlemmer, Ricardo Titze de Almeida, Giane Regina Paludo.

**Formal analysis:** Marcela Correa Scalon, Christine Souza Martins, Ricardo Titze de Almeida, Giane Regina Paludo.

**Funding acquisition:** Giane Regina Paludo.

**Investigation:** Marcela Correa Scalon, Christine Souza Martins, Ricardo Titze de Almeida, Giane Regina Paludo.

**Methodology:** Marcela Correa Scalon, Christine Souza Martins, Gabriel Ginani Ferreira, Franciele Schlemmer, Ricardo Titze de Almeida, Giane Regina Paludo.

**Project administration:** Marcela Correa Scalon, Christine Souza Martins.

**Supervision:** Ricardo Titze de Almeida, Giane Regina Paludo.

**Visualization:** Ricardo Titze de Almeida, Giane Regina Paludo.

**Writing – original draft:** Marcela Correa Scalon, Christine Souza Martins.

**Writing – review & editing:** Marcela Correa Scalon, Christine Souza Martins, Ricardo Titze de Almeida, Giane Regina Paludo.

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
