## [Decision Letter · Decision Letter 0]

2 Oct 2022

PONE-D-22-16135miR-20a is upregulated in serum from domestic feline with PKD1 mutationPLOS ONE

Dear Dr. Paludo,

Thank you for submitting your manuscript to PLOS ONE. After careful consideration, we feel that it has merit but does not fully meet PLOS ONE’s publication criteria as it currently stands. Therefore, we invite you to submit a revised version of the manuscript that addresses the points raised during the review process.

In addition to extensively address the points raised by the reviewer 1, please include more details in the Methods. For example, the info about what primer/assays have been used for the miRNA detection is missing and it is mandatory. Moreover, please include graph of real-time miRNA expression results in the paper and not as supplemental file. In the graph  of miRNA expression results how the fold change has been calculated? Respect to what? It is not clear. Please, specify this also in Methods, in addition to the text and figure legends.

We look forward to receiving your revised manuscript.

Kind regards,

Fabio Sallustio, PhD

Academic Editor

PLOS ONE

Journal Requirements:

2. In the ethics statement in the Methods and online submission information, please specify whether cat owners' consent was obtained and (2) what type you obtained (for instance, written or verbal, and if verbal, how it was documented and witnessed). If the need for consent was waived by the ethics committee, please include this information

Reviewers' comments:

Reviewer's Responses to Questions

**Comments to the Author**

1. Is the manuscript technically sound, and do the data support the conclusions?

Reviewer #1: Yes

2. Has the statistical analysis been performed appropriately and rigorously? 

Reviewer #1: Yes

3. Have the authors made all data underlying the findings in their manuscript fully available?

Reviewer #1: Yes

4. Is the manuscript presented in an intelligible fashion and written in standard English?

Reviewer #1: Yes

5. Review Comments to the Author

Reviewer #1: Comments to the Author

microRNAs (miRs) are the most thoroughly studied none coding RNA. miRs could be of important role in regulating genes expression which may affect disease progression, or as a diagnostic and prognostic biomarker of different diseases. miRs are widely studied in many kinds of diseases, such as ADPKD in human and in mice. But the studies of miRs in cats with ADPKD are scarce. In the present study, detection of serum and plasma levels of 5 miRs were performed in ADPKD cats, as well as in control ones. The results showed that serum miR-20a was upregulated and could be a potential biomarker in ADPKD cats. The detection of miRs of serum samples had a high quality than that of plasma. The results may be of value in the studies of ADPDK in cats. However, there some question needs to be clarified.

1.The 5 miRs selected in this study based on a previous study of miRs of kidney specimens from ADPKD cats, detailed description of these 5 miRs association with ADPKD were absent and needs to be added.

2.The role of miR-20a on cystic formation in ADPKD cats was inferred by the author. However, the relation between serum levels of miR-20a and renal cysts formation or different disease stages were absent.

3.The bioinformatics of miR-20a and its target genes have not been fully discussed.

4.As none coding RNAs is species and genus specific, the deduction (line 37-38) “better understanding of relative expression profiles will further enhance the development of novel diagnostic and prognostic biomarkers as well as prompting patient management” is inappropriate.

6. PLOS authors have the option to publish the peer review history of their article (what does this mean?). If published, this will include your full peer review and any attached files.

Reviewer #1: **Yes: **Liangzhong Sun

---

## [Author Response · Author response to Decision Letter 0]

14 Nov 2022

1. Comments: “the info about what primer/assays have been used for the miRNA detection is missing and it is mandatory.” 

Answer: In Materials and methods, Table 1 was added with the complete information about the primer/assays used in the study.

2. Comments: “please include graph of real-time miRNA expression results in the paper and not as supplemental file.” 

Answer: In Materials and methods, Fig 1 and Fig 2 were added.

3. Comments: “In the graph of miRNA expression results how the fold change has been calculated? Respect to what? It is not clear.” 

Answer: In Materials and methods, the complete information about how the fold change was calculated is added in lines 155 – 169 on manuscript with track changes document.

4. Comments: “If applicable, we recommend that you deposit your laboratory protocols in protocols.io to enhance the reproducibility of your results.”

Answer: All protocols were deposited to Protocols.io, and are provided as supplemental files.

5. Comments: “In the ethics statement in the Methods and online submission information, please specify whether cat owners' consent was obtained and (2) what type you obtained (for instance, written or verbal, and if verbal, how it was documented and witnessed). If the need for consent was waived by the ethics committee, please include this information”

Answer: In Materials and methods, the information about the consent form was added, and a copy with the text (in Portuguese and English) is provided as a supplemental file.

6. Comments: “The 5 miRs selected in this study based on a previous study of miRs of kidney specimens from ADPKD cats, detailed description of these 5 miRs association with ADPKD were absent and needs to be added.” 

Answer: In lines 70 – 90 of Introduction, the detailed descriptions of the 5 miRs and its association with ADPK were added.

7. Comments: “The role of miR-20a on cystic formation in ADPKD cats was inferred by the author. However, the relation between serum levels of miR-20a and renal cysts formation or different disease stages were absent.” 

Answer: In Results and discussion, the information about miR-20a serum levels and renal cysts formation was added in lines 230 – 235 on manuscript with track changes document.

8. Comments: “The bioinformatics of miR-20a and its target genes have not been fully discussed.” 

Answer: In Results and discussion, the bioinformatics of miR-20a and its target genes was added in lines 244 – 249 on manuscript with track changes document.

9. Comments: “As none coding RNAs is species and genus specific, the deduction (line 37-38) “better understanding of relative expression profiles will further enhance the development of novel diagnostic and prognostic biomarkers as well as prompting patient management” is inappropriate.” 

Answer: In Abstract, the sentence was removed.

10. Comments: “While revising your submission, please upload your figure files to the Preflight Analysis and Conversion Engine (PACE) digital diagnostic tool, https://pacev2.apexcovantage.com/.”

Answer: All figures were uploaded to PACE.

---

## [Decision Letter · Decision Letter 1]

6 Dec 2022

miR-20a is upregulated in serum from domestic feline with PKD1 mutation

PONE-D-22-16135R1

Dear Dr. Paludo,

We’re pleased to inform you that your manuscript has been judged scientifically suitable for publication and will be formally accepted for publication once it meets all outstanding technical requirements.

Kind regards,

Fabio Sallustio, PhD

Academic Editor

PLOS ONE

Additional Editor Comments (optional):

Reviewers' comments:

Reviewer's Responses to Questions

**Comments to the Author**

1. If the authors have adequately addressed your comments raised in a previous round of review and you feel that this manuscript is now acceptable for publication, you may indicate that here to bypass the “Comments to the Author” section, enter your conflict of interest statement in the “Confidential to Editor” section, and submit your "Accept" recommendation.

Reviewer #1: All comments have been addressed

2. Is the manuscript technically sound, and do the data support the conclusions?

Reviewer #1: Yes

3. Has the statistical analysis been performed appropriately and rigorously? 

Reviewer #1: Yes

4. Have the authors made all data underlying the findings in their manuscript fully available?

Reviewer #1: Yes

5. Is the manuscript presented in an intelligible fashion and written in standard English?

Reviewer #1: Yes

6. Review Comments to the Author

Reviewer #1: The authors have addressed our concerns and made the manuscript acceptable for publication. In Materials and methods, the information about the consent form was added,If applicable, we recommend that you submit the ethics Ethical review proof as well as.

7. PLOS authors have the option to publish the peer review history of their article (what does this mean?). If published, this will include your full peer review and any attached files.

Reviewer #1: No

---

## [Editor Report · Acceptance letter]

12 Dec 2022

PONE-D-22-16135R1 

miR-20a is upregulated in serum from domestic feline with *PKD1* mutation 

Dear Dr. Paludo:

I'm pleased to inform you that your manuscript has been deemed suitable for publication in PLOS ONE. Congratulations! Your manuscript is now with our production department. 

Kind regards, 

on behalf of

Dr. Fabio Sallustio 

Academic Editor

PLOS ONE